# Classification of Benign and Malignant Renal Tumors Based on CT Scans and Clinical Data Using Machine Learning Methods

Jie Xu [1,*], Xing He [1], Wei Shao [2], Jiang Bian [1] and Russell Terry [3]

1 Department of Health Outcomes & Biomedical Informatics, University of Florida, Gainesville, FL 32611, USA; hexing@ufl.edu (X.H.); bianjiang@ufl.edu (J.B.)
2 Department of Medicine, University of Florida, Gainesville, FL 32611, USA; wei.shao@medicine.ufl.edu
3 Department of Urology, University of Florida, Gainesville, FL 32611, USA; russell.terry@urology.ufl.edu
* Correspondence: xujie@ufl.edu

**Abstract:** Up to 20% of renal masses ≤4 cm is found to be benign at the time of surgical excision, raising concern for overtreatment. However, the risk of malignancy is currently unable to be accurately predicted prior to surgery using imaging alone. The objective of this study is to propose a machine learning (ML) framework for pre-operative renal tumor classification using readily available clinical and CT imaging data. We tested both traditional ML methods (i.e., XGBoost, random forest (RF)) and deep learning (DL) methods (i.e., multilayer perceptron (MLP), 3D convolutional neural network (3DCNN)) to build the classification model. We discovered that the combination of clinical and radiomics features produced the best results (i.e., AUC [95% CI] of 0.719 [0.712–0.726], a precision [95% CI] of 0.976 [0.975–0.978], a recall [95% CI] of 0.683 [0.675–0.691], and a specificity [95% CI] of 0.827 [0.817–0.837]). Our analysis revealed that employing ML models with CT scans and clinical data holds promise for classifying the risk of renal malignancy. Future work should focus on externally validating the proposed model and features to better support clinical decision-making in renal cancer diagnosis.

**Keywords:** renal cancer; classification; CT scans; machine learning





## 1. Introduction

Renal tumors, also known as kidney tumors, are abnormal growths that originate within the renal tissue of one or both kidneys [1]. These tumors exhibit a wide spectrum of behavior, ranging from benign (noncancerous) to malignant (cancerous) growths. The clinical presentation of renal tumors can vary, with some patients remaining asymptomatic, while others may experience a range of symptoms, including flank or lower back pain, hematuria (blood in the urine), unexplained weight loss, and fatigue [2]. Surgical intervention is the cornerstone of treatment for renal tumors, particularly for those that are malignant or symptomatic. Surgical approaches may include partial nephrectomy, where only the tumor and a small margin of healthy tissue are removed, or radical nephrectomy, where the entire affected kidney is removed [3]. Despite the effectiveness of surgery in managing renal tumors, there are growing concerns about the potential overtreatment of small renal masses [4,5]. Studies have revealed that a significant proportion of small renal masses, particularly those with a diameter of 4 cm or less, are benign lesions that may not pose a threat to the patient's health [6,7]. Surgical removal of these benign tumors may expose patients to unnecessary risks and complications associated with surgery. Therefore, there is a critical need for accurate classification of renal tumors using non-invasive techniques to guide treatment decisions, enabling healthcare professionals to prevent unnecessary surgery or other interventions and provide appropriate and effective care to patients [8].

Renal tumors are commonly diagnosed and categorized with the aid of computerized tomography (CT) or magnetic resonance imaging (MRI) scans [9], which offer intricate visualizations of the kidney. However, manually reading and annotating CT scans can be a challenging and time-consuming task. In recent years, machine learning (ML) algorithms

have shown promise in improving the efficiency and accuracy of renal tumor classification based on CT imaging data [10–19]. Researchers have investigated texture features in CT to differentiate renal tumors, with studies employing various texture predictors to improve diagnostic accuracy [10–13].

For example, Deng et al. [10] conducted a study to explore the potential of computed tomography texture analysis (CTTA) in identifying visually imperceptible differences between benign and malignant renal tumors. The study focused on filtration histogram-based parameters, such as entropy and skewness. The researchers discovered that entropy was a useful parameter for differentiating renal cell carcinoma (RCC) from lipid-poor angiomyolipoma, as well as chromophobe RCC from oncocytoma. Logistic regression analysis revealed that both entropy and skewness at fine spatial filter (SSF2) were significant parameters for distinguishing benign and malignant renal tumors. Specifically, an entropy value greater than 5.62 demonstrated high specificity (85.7%) but low sensitivity (31.3%) for predicting RCCs. However, the area under the receiver operating characteristic (ROC) curve (AUC) was only 0.57 for entropy and 0.622 for skewness at SSF2, suggesting that the diagnostic capability of CTTA alone may be limited. In another study, Johannes et al. [11] employed radiomics and machine learning (ML) techniques to differentiate malignant and benign renal masses on CT images. The study found that the random forest (RF) algorithm achieved a significantly higher AUC of 0.83 compared to radiologists, who had an AUC of 0.68. However, this study had a relatively small sample size, as it was based on data from only 94 patients. Peng et al. [12] evaluated the performance of radiomics models based on CT imaging data for the diagnosis of renal cancer. The results showed promise for augmenting radiological diagnosis, particularly for differentiating clear cell RCC (ccRCC) from non-ccRCC, with the highest AUC reaching 0.909. Varghese et al. [13] investigated the diagnostic accuracy of CT-based texture features, such as entropy, mean, and uniformity, in distinguishing renal mass subtypes. The overall contrast-enhanced computed tomography (CECT)-based tumor texture model achieved an AUC of 0.87 ($p < 0.05$) for differentiating benign from malignant renal masses. The potential of ML algorithms, particularly those focused on texture features, lies in improving renal tumor classification and enhancing diagnostic accuracy, which could lead to increased efficiency in diagnostics, reduced burden on radiologists, and minimized inter-observer variability. Moreover, improved diagnostic accuracy and tumor classification using imaging techniques would improve clinicians' ability to appropriately select patients for surgery versus active surveillance and potentially decrease the number of surgeries which are performed for non-malignant renal tumors.

In addition to hand-crafted texture analysis, deep learning (DL) models, such as convolutional neural networks (CNNs), have emerged as powerful tools for renal tumor classification based on imaging data [14–19]. These models have shown promise in enhancing the diagnostic accuracy of renal tumor classification, which could ultimately lead to better patient outcomes and more targeted treatment strategies. For instance, Oberai et al. [14] developed a CNN-based classifier that utilized multiphase contrast-enhanced CT images for renal tumor classification. The classifier achieved an area under the receiver operating characteristic curve (AUC) of 0.82, demonstrating its potential for accurate tumor classification. Similarly, Zabihollahy et al. [15] explored a DL-based method for the automated classification of RCC from benign solid renal masses. Their semi-automated algorithm, which used CECT images, achieved an AUC of up to 0.67. Tanaka et al. [16] employed the Inception-v3 CNN model [20] for determining whether small solid renal masses were benign or malignant. The model achieved the highest AUC of 0.846 with corticomedullary phase (CMP) images, while the lowest AUC of 0.494 was observed with excretory phase (EP) images. In another study, Uhm et al. [17] developed an end-to-end DL model for multiphase CT imaging that accurately differentiated between five major histologic subtypes of renal tumors, including both benign and malignant tumors. The model achieved an impressive AUC of 0.889, outperforming radiologists for most of the subtypes. However, the study was limited to patients with three or more CT phases. Han et al. [18] applied a DL neural network to classify RCC subtypes using biopsy results as labels. The method

achieved an AUC of 0.9, with images acquired at three phases. Zhou et al. [19] investigated the effect of transfer learning on CT images for benign and malignant renal tumor classification. By cross-training the InceptionV3 model pretrained on the ImageNet dataset, they achieved better performance at the patient level (accuracy increased by 2–5%) compared to image-level models (0.69 accuracy). Despite these promising results, most studies relied on multi-phase CT images. While multiphase CT imaging offers more detailed insights into bodily tissues and structures, it also presents drawbacks, such as increased radiation exposure, longer scanning time, higher costs, and potential unsuitability for patients with poor kidney function due to the need for contrast agent clearance [21]. Predicting malignancy risk before surgery using single-phase CT imaging remains a challenge. Roussel et al. [22] conducted a comprehensive review of noninvasive imaging-based tools for characterizing solid renal masses, highlighting their strengths and limitations.

In contrast to existing studies that primarily focus on either hand-crafted texture features using traditional ML methods or multi-phase CT images using DL models, this paper takes a more comprehensive approach by examining contrast enhanced CT images and evaluating both traditional ML and DL models. Our objective is to propose a method for pre-operative renal tumor classification that leverages readily available clinical and CT imaging data. In our investigation, we explored various feature combinations and compared the performance of traditional ML methods, such as XGBoost and RF, with that of DL methods, such as Multilayer Perceptron (MLP) and 3D convolutional neural network (3DCNN). Our findings indicate that the best results were obtained when combining structured clinical data, such as patient demographics and medical history, with radiomics features extracted from CT images. By integrating multiple data modalities, we can substantially improve renal tumor classification, leading to increased accuracy and the development of more effective diagnostic strategies. Future work should externally validate the proposed model and features to further refine and enhance clinical support for renal cancer diagnosis.

## 2. Materials and Methods

### 2.1. Data Source and Study Population

This study was conducted using data obtained from the KiTS21 cohort, which consisted of a total of 300 patients who underwent either partial or radical nephrectomy for radiographically detected renal tumor between 2010 and 2020 [23]. Clinical attributes and imaging data, including patient demographics and comorbidity information, as well as preoperative CT scans before surgery were used to build the classification models to classify malignant and benign tumors. Examples of CT scans can be found in the Supplementary Materials (Figure S1).

### 2.2. Data Processing

#### 2.2.1. Clinical Attributes

The clinical attributes were grouped into three main categories: demographics, vital signs, and comorbidities. Demographic information included the patient's age at the time of nephrectomy and gender. Age was treated as a continuous variable, while gender was treated as a binary variable (male or female). Vital signs encompassed key health indicators, such as body mass index (BMI) and tobacco use status. BMI, which is a measure of body fat based on height and weight, was treated as a continuous variable. Tobacco use status was treated as a binary variable, with patients classified as either tobacco users or non-users. We also considered the presence of comorbidities, which are additional medical conditions that coexist with the primary condition for which the patient is undergoing nephrectomy. Specific comorbidities included in our analysis were myocardial infarction (heart attack) and chronic obstructive pulmonary disease (COPD), among others. Each comorbidity was treated as a binary variable, indicating whether the patient had a history of the condition (yes or no). To provide a comprehensive overview of the patient cohort, we calculated the prevalence of each comorbidity within the study population.

### 2.2.2. CT Scans

For CT scans, we performed a series of image processing and feature extraction steps to prepare the data for analysis using both traditional ML methods and DL techniques. The processing steps involved include image segmentation, resampling, normalization, region of interest (ROI) extraction, radiomic feature extraction, and DL model input preparation.

We utilized voxel-wise majority voting [19] to perform image segmentation on the CT scans. This approach involves combining the results of multiple segmentations to create a final consensus segmentation, which is expected to be more accurate and reliable. To ensure consistency across all CT scans, we resampled the images to achieve a common voxel spacing of 0.78126 mm $\times$ 0.78125 mm $\times$ 0.78125 mm. This step allowed us to standardize the resolution of the images for subsequent analysis. To correct for variations in image intensity that may arise due to differences in scanner settings, we applied min–max normalization to the CT scans. The intensity values were scaled such that the minimum and maximum normalized values corresponded to the 5th and 95th percentiles of intensity, respectively. For each patient, we identified the region of interest (ROI) containing the largest tumor. The ROI was then cropped from the original CT scan for further analysis. The size of the cropped image patch was determined based on the dimensions of the largest tumor observed in the entire patient cohort.

For traditional ML methods, we used the *PyRadiomics* open-source platform [24] to extract radiomic features from the cropped image patches. Radiomic features are quantitative descriptors that capture phenotypic characteristics of tumors and their surrounding tissue. The extracted features were grouped into several categories, including first-order statistics (19 features), shape-based 3D (16 features), shape-based 2D (10 features), gray level cooccurrence matrix (24 features), gray level run length matrix (16 features), gray level size zone matrix (16 features), neighboring gray tone difference matrix (5 features), and gray level dependence matrix (14 features). For input into the DL model, we normalized the ROIs using a commonly used threshold range of $-1000$ to 400. This step helped to focus on the relevant tissue structures while reducing noise and artifacts. We then downsampled the image patches to a shape of $128 \times 128 \times 64$ voxels to ensure compatibility with the input requirements of the DL model.

### 2.3. Machine Learning Algorithms

We developed classification models to categorize patient outcomes using a variety of algorithms, including both ML and DL methods. Specifically, we employed XGBoost and RF as ML methods and MLP and 3DCNN as DL methods. XGBoost (eXtreme Gradient Boosting) is a powerful gradient boosting algorithm known for its high performance in classification tasks [25]. For our analysis, we selected a maximum tree depth of 6, logistic regression as the objective, and employed a heuristic approach to select the fastest tree method, while default values were used for the other parameters. RF is an ensemble learning method that constructs multiple decision trees and combines their predictions for improved accuracy and stability [26]. For both the XGBoost and RF models, we experimented with three different input configurations: (1) clinical attributes, (2) radiomic features, which are extracted from using *pyradiomics* packages, and (3) a combination of clinical and radiomic features.

The MLP, also known as a feedforward neural network, consisted of multiple fully connected layers [27]. The MLP architecture was also used to process the clinical and radiomic features, and its design included input, hidden, and output layers. The 3DCNN architecture was designed to process volumetric image data and consisted of four convolutional layers, each followed by a max pooling layer and a batch normalization layer. The number of filters in the convolutional layers increased in the order of 8, 8, 16, 32, while the kernel size remained fixed at (3,3,3) for all layers. The image patches with dimensions of $128 \times 128 \times 64$ were used as input to the network. Figure 1 provides a visual illustration of the entire classification model development process.

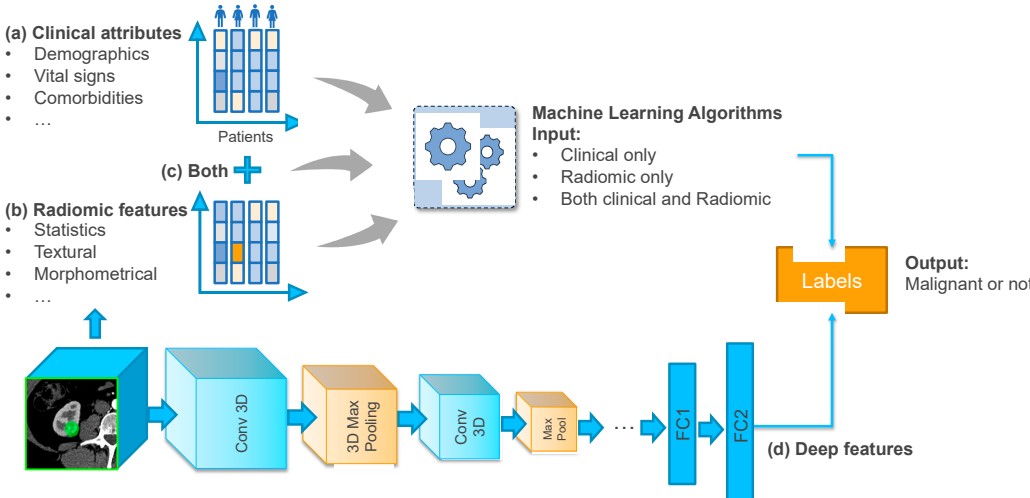

**Figure 1.** Illustration of the prediction framework using different inputs: (**a**) clinical attributes; (**b**) radiomic features; (**c**) combination of clinical and radiomic features; (**d**) DL models.

*2.4. Experiments and Evaluation*

The dataset was divided into a training set and a test set using random sampling, with 80% of the data allocated for training and 20% reserved for testing. The training set was used to train and optimize the ML and DL models, while the test set was used to assess the models' performance on unseen data. To optimize the ML model, we employed a five-fold cross-validation approach on the training set, while the hyperparameters were fine-tuned using a randomized search. The classification performance of the models was assessed using several evaluation metrics, including AUC, precision, recall, and specificity [28]. These metrics provide different perspectives on the models' ability to accurately classify patient outcomes and minimize errors. To ensure robust performance estimates and account for potential variability in the results, we performed bootstrapping with 1000 iterations [29]. This approach allowed us to provide a 95% confidence interval (CI) for each evaluation metric, giving a more comprehensive understanding of the model's performance and its potential generalizability to new, unseen data. We also employed the Kruskal–Wallis test for comparing AUCs across different models for each input, and the Mann–Whitney U test for comparing AUCs between two models [30].

**3. Results**

In our study, we investigated a cohort of 300 patients who underwent either partial or radical nephrectomy due to radiographically detected renal tumor. Out of the entire patient population, 275 (91.7%) were diagnosed with malignant tumors, while the remaining 25 (8.3%) patients were found to have benign tumors. Median radiographic tumor size was 4.1 cm. Table 1 provides an overview of the key characteristics of the study cohort. Interestingly, patients with benign tumors tended to be relatively older than those with malignant tumors, with a mean age of 60.8 years versus 58.7 years, respectively. In terms of gender distribution, the data show that a higher proportion of patients with malignant tumors were male, with 62.2% (171 out of 275) of the total number of patients with malignant tumors being male. The group of patients with malignant renal tumors had a higher percentage of individuals who were current smokers and reported consuming more than two alcoholic drinks daily compared to the group with benign renal tumors. Overall, the data presented in Table 1 shed light on some of the key demographic vital signs of the study cohort, providing a valuable foundation for further analysis and interpretation.

The distribution of the clinical attributes of benign and malignant patients are shown in Figure 2. Among all the clinical attributes, uncomplicated diabetes mellitus and localized solid tumor are the most common ones (20.7% of patients in malignant group and 4% of patients in benign group have uncomplicated diabetes mellitus, and 14.5% patients in

malignant group and 12% of patients in benign group have a localized solid tumor). For most of the clinical attributes except malignant lymphoma and moderate to severe liver disease, more patients in the malignant group were diagnosed.

**Table 1.** Characteristics of the study cohort.

| Characteristics | Total (N = 300) | Benign (N = 25) | Malignant (N = 275) | * p-Value |
|---|---|---|---|---|
| Demographics | | | | |
| * Age, mean (std) | 58.9 (13.8) | 60.8 (15.7) | 58.7 (13.6) | 0.158 |
| Female, N (%) | 120 (40.0) | 16 (64.0) | 104 (37.8) | 0.011 |
| Vital Signs | | | | |
| Body Mass Index, mean (std) | 30.9 (6.7) | 31.1 (6.6) | 30.8 (6.7) | 0.717 |
| Smoking History, N (%) | | | | |
| Current Smoker | 43 (14.3) | 2 (8.0) | 41 (14.9) | 0.345 |
| Never Smoked | 137 (45.7) | 15 (60.0) | 122 (44.4) | 0.133 |
| Tobacco Use, N (%) | | | | |
| Never or not in last 3 months | 295 (98.3) | 25 (100.0) | 270 (98.2) | |
| Alcohol Use, N (%) | | | | |
| More Than Two Daily | 17 (5.7) | 0 (0.0) | 17 (6.2) | |
| Never or not in Last 3 moths | 131 (43.7) | 14 (56.0) | 117 (42.5) | 0.194 |
| Quit in Last 3 months | 1 (0.3) | 0 (0.0) | 1 (0.4) | |
| Tumor Histologic Subtype, N (%) | | | | |
| Angiomyolipoma | 5 (1.7) | 5 (20.0) | 0 | |
| Chromophobe | 27 (9.0) | 0 | 27 (9.8) | |
| Clear cell papillary RCC | 7 (2.3) | 0 | 7 (2.5) | |
| Clear cell RCC | 204 (68.0) | 0 | 204 (74.2) | |
| Collecting duct undefined | 1 (0.3) | 0 | 1 (0.4) | |
| * MEST | 3 (1.0) | 3 (12.0) | 0 | |
| Multilocular cystic RCC | 1 (0.3) | 0 | 1 (0.4) | |
| Oncocytoma | 16 (5.3) | 16 (64.0) | 0 | |
| Papillary | 28 (9.3) | 0 | 28 (10.2) | |
| RCC unclassified | 2 (0.7) | 0 | 2 (0.7) | |
| Spindle cell neoplasm | 1 (0.3) | 1 (4.0) | 0 | |
| Urothelial carcinoma | 3 (1.0) | 0 | 3 (1.1) | |
| Wilms tumor | 1 (0.3) | 0 | 1 (0.4) | |
| Other | 1 (0.3) | 0 | 1 (0.4) | |

* Age: age at nephrectomy; * MEST: mixed epithelial and stromal tumor; * p-value: Mann–Whitney U Test for continuous variables, chi-square test for categorical variables.

Table 2 provides a detailed overview of the performance of all prediction models evaluated in this study. Among the models tested, the combination of clinical and radiomic features yielded the best overall performance, with an AUC [95% CI] of 0.719 [0.712–0.726], a precision [95% CI] of 0.976 [0.975–0.978], a recall [95% CI] of 0.683 [0.675–0.691], and a specificity [95% CI] of 0.827 [0.817–0.837]. The results of the Kruskal–Wallis Test indicate that the models' performance is significantly different for each input. Additional information, including the results of the Mann–Whitney U test comparing AUCs between two models, can be found in the Supplementary Materials (Table S1). An interesting observation from the study was that the traditional ML model, RF, which utilized the combination of clinical and radiomic features as input, performed slightly better in terms of precision compared to the DL model. Specifically, the RF model achieved a precision [95% CI] of 0.976 [0.975–0.978], while the DL model achieved a precision [95% CI] of 0.958 [0.956–0.960]. The difference in performance may be explained by the relatively small dataset of CT scans used to train the DL model, which could have limited the model's ability to effectively identify and learn patterns within the data. It is also important to note that most related DL-based studies that focus on classifying benign and malignant renal tumors have utilized multi-phase CT images. In contrast, our study relied on single-phase CT images, which may present different challenges and opportunities for model develop-

ment. Overall, these findings underscore the value of integrating both clinical and radiomic features for predicting renal tumor diagnosis. The successful combination of these features has the potential to enhance the accuracy of tumor classification. However, further research is needed to validate these findings, particularly in larger and more diverse patient cohorts.

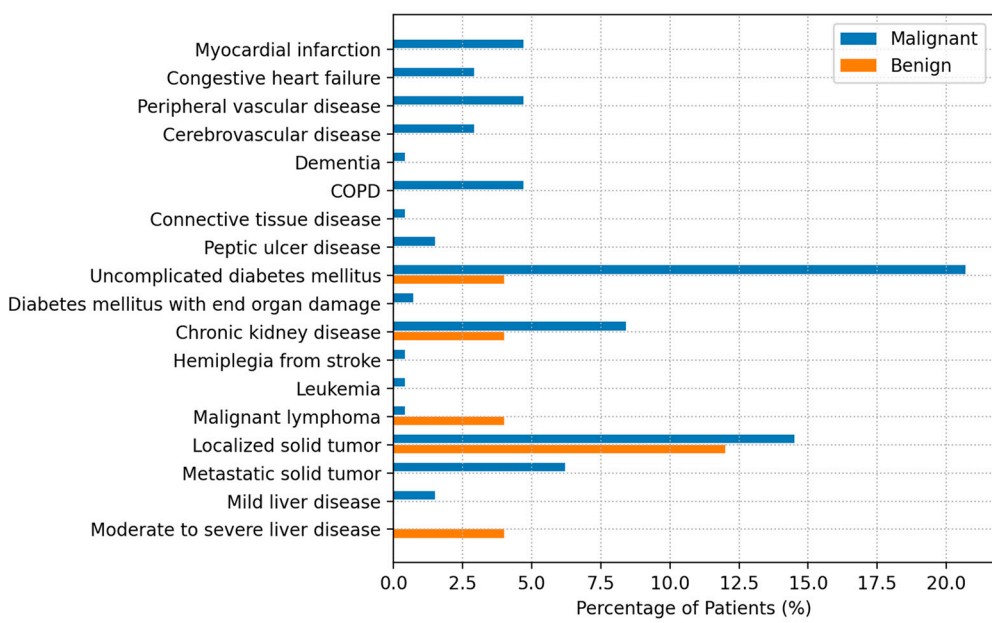

**Figure 2.** Clinical attributes of benign and malignant patients. COPD = chronic obstructive pulmonary disease.

**Table 2.** The performance of all prediction models.

| Input | Methods | Metrics | AUC | Precision | Recall | Specificity | * p-Value |
|---|---|---|---|---|---|---|---|
| Clinical | * MLP | mean (std)<br>95% CI | 0.705 (0.117)<br>(0.698, 0.712) | 0.973 (0.028)<br>(0.972, 0.975) | 0.654 (0.160)<br>(0.645, 0.664) | 0.816 (0.178)<br>(0.805, 0.827) | <0.001 |
| | XGBoost | mean (std)<br>95% CI | 0.639 (0.108)<br>(0.632, 0.646) | 0.972 (0.032)<br>(0.970, 0.974) | 0.561 (0.140)<br>(0.553, 0.57) | 0.837 (0.170)<br>(0.827, 0.848) | |
| | * RF | mean (std)<br>95% CI | 0.651 (0.114)<br>(0.644, 0.658) | 0.971 (0.029)<br>(0.969, 0.972) | 0.606 (0.126)<br>(0.598, 0.614) | 0.816 (0.169)<br>(0.805, 0.826) | |
| CT Scans | 3DCNN | mean (std)<br>95% CI | 0.601 (0.141)<br>(0.592, 0.610) | 0.958 (0.037)<br>(0.956, 0.960) | 0.582 (0.176)<br>(0.571, 0.593) | 0.736 (0.208)<br>(0.723, 0.749) | |
| Radiomic | MLP | mean (std)<br>95% CI | 0.581 (0.135)<br>(0.572, 0.589) | 0.948 (0.05)<br>(0.945, 0.951) | 0.575 (0.217)<br>(0.561, 0.588) | 0.676 (0.220)<br>(0.662, 0.690) | <0.001 |
| | XGBoost | mean (std)<br>95% CI | 0.670 (0.133)<br>(0.662, 0.678) | 0.970 (0.026)<br>(0.968, 0.971) | 0.666 (0.158)<br>(0.656, 0.676) | 0.772 (0.183)<br>(0.761, 0.784) | |
| | RF | mean (std)<br>95% CI | 0.700 (0.116)<br>(0.893, 0.707) | 0.955 (0.039)<br>(0.953, 0.957) | 0.665 (0.132)<br>(0.657, 0.673) | 0.805 (0.164)<br>(0.794, 0.815) | |
| * Both | MLP | mean (std)<br>95% CI | 0.584 (0.137)<br>(0.575, 0.592) | 0.951 (0.037)<br>(0.949, 0.953) | 0.586 (0.208)<br>(0.573, 0.599) | 0.686 (0.209)<br>(0.673, 0.699) | <0.001 |
| | XGBoost | mean (std)<br>95% CI | 0.718 (0.111)<br>(0.711, 0.725) | 0.972 (0.025)<br>(0.970, 0.974) | 0.673 (0.149)<br>(0.663, 0.682) | 0.800 (0.159)<br>(0.790, 0.810) | |
| | RF | mean (std)<br>95% CI | 0.719 (0.116)<br>(0.712, 0.726) | 0.976 (0.024)<br>(0.975, 0.978) | 0.683 (0.132)<br>(0.675, 0.691) | 0.827 (0.163)<br>(0.817, 0.837) | |

* MLP: multilayer perceptron; * RF: random forest; * Both: clinical and radiomic features; * p-value: Kruskal–Wallis Test.

## 4. Discussion

In our study, we analyzed a cohort of 300 patients, the majority of whom had malignant tumors, accounting for 91.67% (275/300) of the total cases. This high prevalence of malignant tumors is not surprising, given that this is a restricted cohort [19], and all patients

underwent either partial or radical nephrectomy due to suspected renal malignancy. We observed notable differences (*p*-value < 0.05) in gender among patients with malignant tumors, with a higher incidence of malignant renal tumors found among male patients. This relationship has been described previously, as male sex is noted by the American Urological Association's Renal Mass Guideline to be one of the most reliable predictors of malignancy along with tumor size [31]. The precise reasons for this gender difference in renal tumor incidence remain unclear, and further research is needed to elucidate the underlying mechanisms and risk factors.

Our study primarily focuses on differentiating between benign and malignant renal tumors. By integrating information from both clinical attributes and pre-operative CT scans into our model, we achieved better performance compared to using imaging data alone. This approach leverages the complementary insights provided by clinical and imaging data, offering a more comprehensive understanding of factors related to malignancy risk. Combining these data sources leads to improved diagnostic accuracy and more effective strategies for renal tumor classification, emphasizing the importance of considering multiple data modalities when developing diagnostic tools and highlighting the potential benefits of integrating clinical and imaging data in renal tumor management [32,33].

The fact that all patients in our study underwent either partial or radical nephrectomy underscores the complexity of accurately classifying patients with malignant and benign renal tumors. The recommendation for nephrectomy even in patients with benign renal tumors suggests that their clinical presentation and imaging findings closely resembled those of patients with malignant tumors. This phenotypic similarity highlights the difficulty in accurately differentiating between malignant and benign renal tumors when relying solely on pre-operative assessments. Improving pre-operative classification methods is crucial to reducing unnecessary surgeries and providing better patient outcomes by developing more accurate and reliable tools for preoperative differentiation between malignant and benign renal tumors.

Our study has several limitations. First, we relied exclusively on a public dataset of only 300 cases, of whom 91.67% were malignant. This is an imbalanced dataset due to over-representation of malignant tumor types compared to the expected rate of approximately 80% in the general population with tumors in this size range. Although we employed data augmentation while training the 3DCNN model, incorporating additional scans and cases in future studies, particularly for those with benign tumor types, could help improve the model's performance. Furthermore, using an independent dataset for validation would enhance the reliability of our findings. Another limitation of this work is that it does not include tumor types that have recently been added to the WHO classification of renal tumors (e.g., low-grade oncocytic tumors). Future radiomics work should consider the inclusion of such tumor types, which are not as common as the included tumors, but may be meaningful for the performance of predictive models, improving patient care. Third, because these data were published as part of a tumor segmentation task challenge, only a limited range of clinical attributes are available. Future research should incorporate more diverse clinical data modalities, such as clinical notes, genetic data, and additional clinical attributes, to provide a more comprehensive understanding of the factors affecting renal tumor classification. Fourth, our study tested a limited number of methods, specifically RF, XGBoost, MLP, and a 3DCNN model. As more extensive datasets become available, future work could explore a wider range of ML methods to enhance the model's performance and accuracy.

Moreover, in addition to differentiating between benign and malignant renal tumors, identifying specific subtypes of renal tumors, such as clear cell, papillary, or chromophobe renal cell carcinoma, is also of paramount importance [13,18]. These subtypes exhibit distinct clinical characteristics and may require tailored treatment and surveillance strategies. Therefore, accurate identification of tumor subtypes is crucial for guiding personalized treatment plans for patients with renal tumors. To develop ML algorithms capable of differentiating between benign and malignant lesions and identifying tumor subtypes, a large dataset of annotated CT scans is typically required. This dataset should encompass a

diverse range of lesions and subtypes to ensure that the algorithm can generalize effectively to new cases [33]. Moreover, it is essential to rigorously evaluate the algorithm's performance using a validation dataset to confirm its accuracy in classifying lesions and subtypes. Future research will focus on identifying specific subtypes utilizing more extensive datasets and advanced ML techniques.

## 5. Conclusions

Our study aimed to develop a method for pre-operative renal tumor classification using readily available structured clinical and CT imaging data. We tested both traditional ML and DL methods to create the classification models. Our findings revealed that the integration of clinical and radiomics features yielded the most robust performance, underscoring the potential of ML and DL models in conjunction with CT scans and clinical data for accurately classifying renal tumors and assessing malignancy risk. Future research should concentrate on external validation of the proposed model and features, while also examining the application of these models in clinical settings. Additionally, the integration of supplementary data sources, such as clinical notes and genomics data, should be considered for enhanced classification performance.

**Supplementary Materials:** The following supporting information can be downloaded at: https://www.mdpi.com/article/10.3390/informatics10030055/s1, Figure S1: Examples of CT scans; Table S1: P-values from the Mann–Whitney U test comparing AUCs of two models.

**Author Contributions:** Conceptualization, J.X., J.B. and R.T.; methodology, J.X.; validation, J.X. and R.T.; formal analysis, J.X. and R.T.; investigation, J.X. and X.H.; writing—original draft preparation, J.X.; writing—review and editing, X.H., W.S., J.B. and R.T.; visualization, J.X.; supervision, J.B. and R.T. All authors have read and agreed to the published version of the manuscript.

**Funding:** This research received no external funding.

**Institutional Review Board Statement:** Not applicable.

**Informed Consent Statement:** Not applicable.

**Data Availability Statement:** Data are publicly available at https://github.com/neheller/kits21 (accessed on 28 June 2022).

**Conflicts of Interest:** The authors declare no conflict of interest.

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
