# Peer review of "Classification of Benign and Malignant Renal Tumors Based on CT Scans and Clinical Data Using Machine Learning Methods"

_informatics, doi:10.3390/informatics10030055_

Round 1

Reviewer 1 Report

Thank you for the chance to review your manuscript. An interesting approach for kidney cancer. 

A few suggestions comes to mind.

The approach is merely on kidney cancer. The results are promising but still figures low on sensitivity and specificity for clinical applications <0.84. This from a biomedical approach. Would suggest to that the work to aim as a proposal method, instead as an non-invasive method. As stated in the conclusions.

I congratulate the authors for leaving that dataset as is. Several reports on clinical applications propose to artificial increase dataset for size or balance purposes without considering the clinical responsibility of such decision. 

Well written manuscript

Reviewer 2 Report

The paper applies machine learning methods to classify benign and malignant tumors based on CT data.

The introduction is easy to read and contains appropriate reference to related studies and comparison between different deep learning  or hand crafted models.

The description of the dataset and the characteristics of the population is detailed. 

I would suggest to improve the description of the technical aspect of the XGBoost method. Although this paper focus on clinical aspect of the framework, it is important to describe more in details the type of architecture and loss function used for reproducibility.

The results section contains good results in terms of specificity analysis and p-value. I would add some images of the malignant and benign feature to understand visually how the radiomics feature differs in the image. 

I suggest to improve the paper according to the above suggestions.

Reviewer 3 Report

- I read with interest this manuscript on renal tumours and the use of radiomics and deep learning for the differentiation between benign and malignant tumours. The idea is interesting but not novel and a series of papers have attempted such a differentiation with radiomics and deep learning.

-The major problem I see with this manuscript is that it uses an outdated classification of renal tumours. The recent version of WHO classification has separated low-grade oncocytic tumours (LOTs) to include them in the benign group. Therefore unless the authors have exact pathology results specifying LOTs and hybrid tumours , the results of this paper are completely outdated and cannot be used.

- a table with the exact pathological type of each tumour from the initial dataset needs to be provided and the authors have to modify the analysis according to the new classification. 

- what is the reason that the authors used non contrast enhanced CT?

- why did the authors not include wavelet transformed features of PyRadiomics? in diverse datasets wavelet transformation allows for the elimination of some of the batch effects while highlighting edges that are important in the characterisation of renal tumour appearance

- Table 1 does not provide any information with regards to the cohort of tumours. Pathological characteristics are essential.

- The authors need to explain the results that make no sense such as the relationship between myocardial infarction, heart failure, cerebrovascular disease with malignant tumours (see figure 2).

- statistical comparison should be performed between the results of all models using a method such as Delong's to compare AUCs

- a checklist like CLEAR needs to be used and all relevant information with regards to radiomics that is missing needs to be added to the text. https://insightsimaging.springeropen.com/articles/10.1186/s13244-023-01415-8

English is ok

Round 2

Reviewer 2 Report

The authors have answers my previous comments incorporating new examples in the supplementary materials. The description of the dataset and the characteristics of the population is detailed. The paper can be accepted in the current state.

Author Response

Thank you!

Reviewer 3 Report

The authors have tried to address my concerns especially with regards to the tumour classification. They added the list of pathological diagnoses and tried to explain the fact that they do not have other types of tumours such as LOTs. However, this has not been explained at all inside the text of the manuscript. The authors need to clearly acknowledge this limitation in the text so that the readers are aware of this important shortcoming. I suggest that a phrase is added in the limitations section of the discussion saying something like this:

"A limitation of this work is that it does not contain tumour types that have not been recently added to the WHO classification of renal tumours (e.g. low-grade oncocytic tumours - LOTs). Future radiomics work should consider the inclusion of such tumour types which are not as common as the ones included, but may be of significance for the performance of predictive models, improving patient treatment".

If this is addressed the rest is ok for publication

Thorough check for spelling needs to be done

Author Response

Thank you! We have added the limitation to the DISCUSSION section. Specifically, we added the following statement to the manuscript:

Another limitation of this work is that it does not include tumor types that have recently been added to the WHO classification of renal tumors (e.g., low-grade oncocytic tumors). Future radiomics work should consider the inclusion of such tumor types, which are not as common as the included tumors but may be meaningful for the performance of predictive models, improving patient care.